# LARGE-SCALE MULTI-AGENT REINFORCEMENT LEARNING FOR TRAFFIC SIGNAL OPTIMIZATION

## ABSTRACT

We present a novel approach to Traffic Signal Control (TSC) in a multi-agent environment by modeling communication among agents as a sequence problem, enabling intersections within road networks to communicate with one another. Taking inspiration from point cloud processing and graph neural networks, we make our architecture capable of handling variable road network topologies, including differing numbers of intersections and intersection types, and demonstrate this by successfully training on real & randomly generated road networks and traffic demands. Furthermore, we demonstrate that even utilizing minimal state information can achieve competitive performance.

## 1 INTRODUCTION

When traffic lights are controlled effectively, the industry, the climate and the individual profits. Traffic congestion causes 3.9B Euro in economic damages due to lost time each year in Germany alone (Inrix, 2022). In stop-and-go traffic, emissions are 29 times higher than in free flowing traffic (Greenlight, 2024).

Due to these impacts, Traffic Signal Control (TSC) has become a crucial field of research. It involves using traffic lights at intersections to manage traffic flow with the objective to reduce congestion and enhance safety. However, these goals are not easily achieved due to the problem's dynamic and unpredictable nature. Traffic flow varies significantly throughout the day, influenced by factors such as rush hours, weather, accidents, events, etc. that require real-time adaptive solutions. Additionally, traffic signals can not be managed as stand-alone agents: intersections serve as nodes in a larger network, and traffic at one intersection will affect the flow at others. Coordinating signals across multiple intersections is thus necessary. Different stakeholders in TSC also have varying objectives that must be balanced. For instance, drivers want minimal waiting time, pedestrians and cyclists prioritize safety, and city planners aim to reduce overall emissions and costs incurred due to delays. Existing traffic infrastructures most commonly employ the usage of deterministic models such as round-robin scheduling or predetermined phase periods which do not account for such objectives (Tomar et al., 2022).

The hardware that a TSC scheme is deployed to also poses a barrier to the wide-spread implementation of more advanced TSC algorithms. In German cities, a fair percentage of the traffic controllers found at intersections cannot be dynamically controlled. In TSC algorithm implementation, these must also be accounted for, e.g. through a one-time change of signal plans which best support the dynamically controlled traffic lights. Faulty traffic sensor data or low sensor coverage further limit applicability, and historical data from sensors may be hard to acquire on a large scale. Thus, an approach that does not require large amounts of real data for training is beneficial.

In this work, we use the SUMO simulation environment (Lopez et al., 2018a) to simulate generated road networks. Our architecture for TSC, covered in section 2, takes lane-level observations such as number of vehicles on the lane and current traffic signal status as input. It then uses a Transformer to allow lanes to

attend to one other and projects their observations into a hidden representation. These are pooled and fed into a fixed-sized MLP, forcing the model to compress the information into a size which is the same for any road network, independent of the number of lanes. This allows our model to be capable of handling changing road networks, as is often the case due to road closures, construction, accidents etc.

Our aim with this work was to address the problems outlined above. We focused on employing the most recent advances in the existing literature, and tackle the problem of TSC in a novel manner. The following are our main contributions:

- We built an automated pipeline of dataset generation. This is a significant contribution, as we have addressed the need of large amounts of data to train the transformer model effectively without reliance on limited real-world data. Our pipeline is not only able to produce varying complexity road networks (from a simple ring network to a large scale city network), but it can also produce complex and dynamic traffic flows.
- We treat inter-agent's spatial dependencies as a 2D sequence problem and utilize the powerful transformer architecture to model this sequence. This approach is our primary novelty as it differs from existing literature which use transformer models to encode the state history Chen et al. (2021).
- Due to our novel modeling approach, our pipeline is capable of handling variable input sizes - both in the number of intersections in a road network and the intersection sizes. This alleviates the problem of fixed input sizes prevalent in other model architectures. Our model can thus be easily transferred and deployed regardless of the training environment setup.
- Finally, with extensive experiments, we show that usage of minimal state information – available using tools such as Google Maps – is sufficient to achieve competitive performance. This contribution serves as an important breakthrough, suggesting that reliance on expensive sensor technology might not be necessary.

## 1.1 RELATED WORKS

Table 1 shows an overview of important RL aspects of the architectures covered in this section.

A well defined evaluation framework can be found in the RESCO benchmark (Ault & Sharon, 2021), which evaluated IDQN, IPPO, MPLight and FMA2C on simulations of varying excerpts of two German cities, Cologne and Ingolstadt. Short summaries of these four methods can be found in the RESCO paper, and one key difference to ours is the network independence we built into the architecture. Additionally, only FMA2C uses a multi-agent reinforcement learning (MARL) approach, while the agents of the other three architectures are independent. MARL is a scalable approach to controlling larger-sized network, while independent agents seem to reach a limit in effectiveness (Shi et al., 2023).

Both IG-RL and MuJAM from Devailly et al. (2022; 2024) are also network-independent, with IG-RL using a deep Q-learning approach similar to IDQN, IPPO and MPLight with vehicles as the nodes of the graph (the state), while MuJAM uses a model-based RL approach of applying a world model for planning to the TSC problem domain.

RGLight (Shi et al., 2023) uses a policy ensemble of graph convolutional networks (GCNs), allowing for a zero-shot transfer to other road networks, as do IG-RL and MuJAM.

A recently published preprint, CityLight (Zeng et al., 2024), comes closest to our architecture, using Multi-Agent PPO as the RL foundation and a network independent representation of observations, which however are at the intersection level.

| Paper | Method | Actions | State | Reward | Benchmarks | Eval. |
|---|---|---|---|---|---|---|
| IG-RL Devailly et al. (2022) | Deep Q-Learning; vehicles, lanes, traffic lights as nodes; zero-shot transfer to large network | Binary hold or switch phase | Demand at the vehicle (speed, position on lane) and lane level (#vehicles, avg. speed), connectivity | Neg. sum of queue lengths | Small synthetic & large scale Manhattan (ca. 4k TSCs) | Change in delay |
| MuJAM Devailly et al. (2024) | Model-based RL, planing by modeling the dynamics of the environment | Select phase | Graph (vehicle positions/speeds & controller states) | Neg. sum of queue lengths | Small synthetic & large scale Manhattan (ca. 4k TSCs) | Change in delay |
| RGLight Shi et al. (2023) | Distributional RL, GCN as policy network. Improvement to zero-shot transfer through policy ensemble | Binary hold or switch phase | Status of controller, connectivity, vehicles and lanes | Neg. sum of queue lengths | Synthetic from IG-RL & Manhattan (75 TSCs, 550 intersections), Luxembourg (22 TSCs, 482 intersections) | Travel time, queue length, delay |
| CityLight Zeng et al. (2024) | Multi-agent PPO, neighborhood representation fusion | Select phase | Vehicle queues for each phase, connectivity | Avg. queue lengths in neighbourhood | Large scale Chinese cities (97 to 13952 TSCs) | throughput, avg. travel time |

Table 1: Related works

## 2 METHODOLOGY

### 2.1 PROBLEM FORMULATION

We model the TSC optimization in a multi-agent environment as an MDP involving agents $i \in \{1, \dots, N\}$ – the intersections and associated traffic signals – where agents can take actions in their respective action space $a^i \in \mathcal{A}$. The joint action space is denoted $\mathcal{A} = \mathcal{A}^i \times \cdots \times \mathcal{A}^N$, and the agents' actions lead to a global reward $r \in \mathcal{R}$. The global state $s \in \mathcal{S}$ of the MDP is assumed to be unknown, instead each agent $i$ has access to a subset of the global state $s^i \in \mathcal{S}^i \subseteq \mathcal{S}$, mainly consisting of observations in its proximity, and

$$\mathcal{S}^i \subseteq \bigcup_{k=1}^{N} \mathcal{S}^k \subseteq \mathcal{S}. \tag{1}$$

The shared policy $\pi_{\phi_\pi}(a^i|s^i)$, determining each agent's action, and the value function $V_{\phi_v}^\pi(s)$ both depend on estimates of the state, which is why robust state estimates are crucial for finding optimal policies and value functions. This motivates the generation of an *enriched agent's state* $\hat{s}^i \in \hat{\mathcal{S}}^i \supseteq \mathcal{S}^i$ by letting agents exchange

state information over a communication channel

$$f_\theta : \mathcal{S}^1 \times \cdots \times \mathcal{S}^N \to \hat{\mathcal{S}}^1 \times \cdots \times \hat{\mathcal{S}}^N \quad (2)$$

which we parameterize by two different neural network architectures in this work. We optimize for $V^\star(\hat{s})$ and $\pi^\star(a^i|\hat{s}^i)$ by sampling state-action traces $\tau$ and jointly maximizing the expected discounted cumulative reward for optimal parameters $\Theta^\star = \{\theta^\star, \phi_\pi^\star, \phi_v^\star\}$ through Proximal Policy Optimization (PPO) Schulman et al. (2017).

### 2.1.1 ACTIONS & REWARDS

**Actions** $a^i$: The action for our agents is to change their traffic light phase, which is selected from the set of all possible phases for the intersection, i.e. $\mathcal{A}^i$ (red/ yellow/ green for traffic flow control). Since the cardinality $|\mathcal{A}^i|$ is not the same for all $i$, we pad the action space of all agents and in case of an invalid phase assignment, the agent remains in the current phase. In order to enhance the learning of the admissible actions, we additionally condition the policy on the action space. We also define minimum phase times $t_{phase}^{min}$ and maximum phase times $t_{phase}^{max}$ to prohibit the agent from getting stuck in local optima.

$$a^i \sim \pi(\cdot|\hat{s}^i, t_{phase}^{min}, t_{phase}^{max}, \mathcal{A}^i) \quad (3)$$

**Reward** $r$: We use difference in vehicle waiting time as the reward function, a commonly used reward function in the literature and the default reward function in our framework. Alegre (2019); Reza et al.

$$r = \sum_{i=1}^{N} W_{t-1}^i - W_t^i \quad (4)$$

### 2.1.2 VARYING STATE INFORMATION

We differentiate between three different sources of state observations, sorted by cost of implementation in a real world scenario: 1. traffic information that the agent has on itself, 2. traffic information that can be gathered from cloud providers, 3. information that requires expensive sensory infrastructure. The first source of information applies agent-wide, while sources 2 & 3 are available on lane-level, which is why we denote agent $i$'s lane count as $L^i$. In our pipeline, agent $i$ can be enabled to have access to three different levels of observing $s^i$, progressively incorporating the sources of state observation from above.

**No Traffic Observation** In this scenario, agent $i$ only receives information related to its own traffic lights. The specific state vector components are $s_{notr}^i = \{$number of traffic light phases, lane position, current phase, min. green signal time, remaining time in current green phase, timer, lane angles, lane max speed, action space, turning options$\}$. Features like $angles$, $position$, and $turning options$ are provided to learn the spatial network setup. The feature $timer$ counts repeatedly from 1 to 100 to help the agents calibrate their behavior, like green waves engineered by city planners.

**Limited Traffic Observation** Here, state information is expanded to include high-level traffic metrics available on platforms like Google Maps, TomTom, Here, or Inrix, without installing local sensors. This setup allows the model to make more informed decisions by understanding the general flow of traffic. The information available to agent $i$ is $s_{lim}^i = \{s_{notr}^i, \{\mu_v^{i,1}, \ldots, \mu_v^{i,L^i}\}\}$, where $\mu_v^{i,k}$ denotes the average speed on agent $i$'s $k$th lane.

**Full Observation** This scenario includes detailed metrics for a thorough representation as enabled by traffic sensors: $s_{full}^i = \{s_{lim}^i, \{\rho^{i,1}, \ldots, \rho^{i,L^i}\}, \{q^{i,1}, \ldots, q^{i,L^i}\}\}$, where $\rho^{i,k}$ and $q^{i,k}$ are the traffic densities and queue lengths on agent $i$'s $k^{th}$ lane.

## 2.2 MODEL DETAILS

### 2.2.1 PERMUTATION-INVARIANT LANE ENCODING

Let us now introduce the first step of our pipeline: the lane encoding mechanism. Each agent $a^i$ has access to information from its incoming and outgoing lanes, denoted as $\{l^{i,1}, \ldots, l^{i,L^i}\}$. The value of $L^i$ varies based on the intersection type, and assigning a canonical order to these lanes is challenging due to the diverse shapes of intersections in a road network. We, therefore, seek to find a canonical encoding of the lane-level information through permutation-invariance and take inspiration from point cloud processing to also minimize the influence of the lance count $L^i$. We concatenate lane-level information individually for each lane and feed it through a PointNet encoder Qi et al. (2017), which consists of a Multi-Layer Perceptron (MLP) projecting lane features to a high-dimensional space and a permutation-invariant reduction by max-pooling over an agent's individual lanes. By projection to a high-dimensional space, the information content can be well retained beyond the max-pooling operation as Qi et al. demonstrated in their seminal work Qi et al. (2017). The weights of the MLP are shared across all agents and an in-depth visualization is shown in Fig. 1.

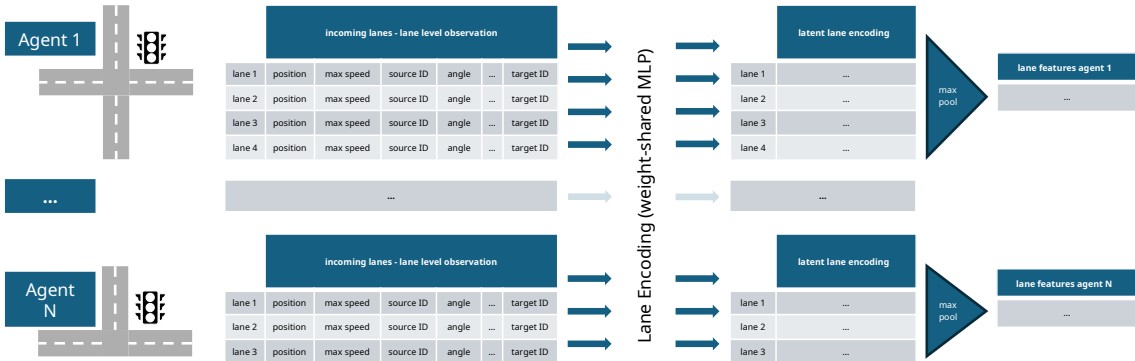

Figure 1: **Permutation-Invariant Lane Encoding:** The PointNet Qi et al. (2017)-inspired lane-encoding concatenates lane-level observations, projects them into a high dimension and creates a permutation-invariant representation through max-pooling. This enables a canonical representation of lane-level observations for each agent.

### 2.2.2 INTER-AGENT COMMUNICATION

As mentioned before, we approach the communication among agents as the primary sequence modeling problem by allowing agent's states $s^i$ to attend to other states $s^k, k \neq i$. We condition our transformer network on the spatial relations between agents by overlaying a 2D positional encoding onto $s^i$, based on normalized longitude and latitude. We can additionally explicitly influence attention values by utilizing an attention mask $\mathbf{M}$ that exponentially decays with distance between individual agents, allowing only attending to agents in close proximity as

$$\mathbf{M} = (m_{i,j}), m_{i,j} = e^{d_{i,j}/C}, \forall d_{i,j} \in \mathbf{D}, C \in \text{const. and } \mathbf{D} \in \mathbb{R}^{N \times N} : \text{the distance matrix.} \quad (5)$$

Once an agent's permutation-invariant representation of its lane observations is obtained, we concatenate it with agent-level observations (sec. 2.1.2) as $s^i$ and feed them to the Transformer to get $\hat{s}^i_{trans}$.

### 2.2.3 MODEL TYPES

Enriched states $\hat{s}^i$ are mapped to action $a^i$ and the value estimate $v$ by dedicated MLPs $\pi_{\phi_\pi}$ and $V_{\phi_v}$ as

$$a^i \sim \pi_{\phi_\pi}(\hat{s}^i), \forall i \in \{1, \ldots, N\}, \quad v = \sum_{i=0}^{N} V_{\phi_v}(\hat{s}^i) \tag{6}$$

and we also create an additional baseline model without any inter-agent communication, which only uses an MLP $f_\theta$ for transformation of $s^i$, before feeding it to the value and policy networks

$$a^i \sim \pi_{\phi_\pi}(f_\theta(s^i)), \forall i \in \{1, \ldots, N\}, \quad v = \sum_{i=0}^{N} V_{\phi_v}(f_\theta(s^i)). \tag{7}$$

This leads us to two different models for which we show the pipeline in Fig. 2:

- **Transformer** for attention-based inter-agent communication.
- **Simple MLP** without any information exchange between agents.

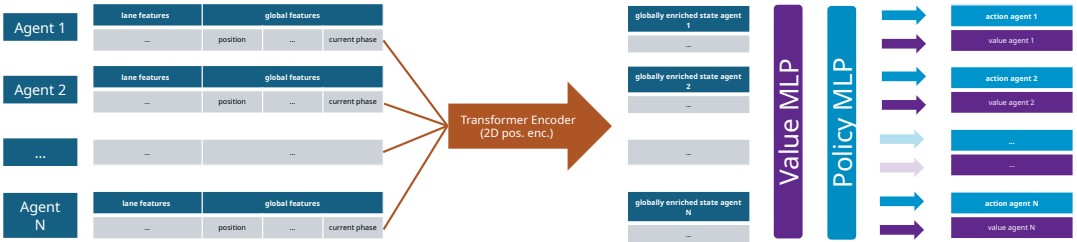

Figure 2: **Final Network Design and Pipeline**: Permutation-invariant lane features are concatenated with agent-level observations to produce $s^i$ and fed through a communication network, after which dedicated networks derive actions and values from the enriched agent's state $\hat{s}^i$.

### 2.3 TRAINING ENVIRONMENT

As detailed in the section 2.4, we use SUMO (Simulation of Urban MObility) for traffic data simulation, where an environment $E$ consists of 1. the road network and 2. the traffic demand. Our model can be trained using either synthetically generated environments or use imported real world road networks (for instance, the map of Zürich). This was built into our pipeline to allow for thorough training and testing of our model performance, and for evaluating its ability to handle a wide range of network complexities.

Our marked contribution here is that we have implemented a methodology to automate the dataset and data distribution generation process, where we can sample new environments conditioned on hyperparameters, such as the allowed range for $N$ (number of agents / intersections) and the average traffic density $\rho$, i.e. $E \sim P_E(\cdot|N, \rho)$. This also allows us to resample the environment during training to avoid over-fitting or to train on several environments simultaneously. Fig. 3 shows some of our generated road network samples for low values of $N$ and a more complicated example is shown in Fig. 6a.

### 2.4 TOOLS & FRAMEWORKS

**Simulation Environment** As mentioned in section 2.3, we use SUMO Lopez et al. (2018b), which has established itself as the standard for simulating traffic environments.

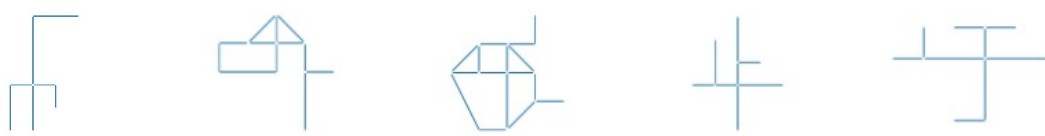

Figure 3: Some examples of randomized road networks with relatively low $N$-values, that we can generate on the fly while training our networks.

**RL Framework** SUMO-RL allows easy access to SUMO's API and creating multi-agent PettingZoo environments Alegre (2019); Terry et al. (2021). These integrate well with Ray's RLlib, which implements state-of-the-art model-free algorithms (PPO, DQN) on distributed systems Liang et al. (2018). PyTorch is used for the implementation of our custom neural networks Paszke et al. (2019).

**Training & Evaluation** Our training runs are deployed using Docker/Apptainer, and we train on ETHZ's Euler cluster. Training progress and results were analysed using WandB.ai Biewald (2020).

## 2.5 TRAINING

As mentioned in section 2.3, we can choose to train either on a single environment or multiple at the same time and the algorithms for doing so are presented in Alg. 1 and Alg. 2, respectively.

---

**Algorithm 1** Single Environment Training

**Require:** $episodes > 0$
**Require:** $B > 0$      ▷ batch size
  $E \sim P_E(E)$
  **while** $n < episodes$ **do**
    $T \leftarrow [\dots]$     ▷ container for traces
    **for** $b \in \{1, \dots, B\}$ **do**
      $\tau \leftarrow step(E)$
      $T \leftarrow append(\tau)$
    **end for**
    $V, \pi, f_\theta \leftarrow PPO(T)$
  **end while**

---

**Algorithm 2** Multiple Environment Training

**Require:** $episodes > 0$
**Require:** $B > 0$
**Require:** $n_E \geq 2$     ▷ simult. environments
  **while** $n < epidodes$ **do**
    $T \leftarrow [\dots]$
    **for** $i \in \{1, \dots, n_E\}$ **do**    ▷ parallelized
      $E \sim P_E(E)$
      **for** $b \in \{1, \dots, B/n_E\}$ **do**
        $\tau \leftarrow step(E)$
        $T \leftarrow append(\tau)$
      **end for**
    **end for**
    $V, \pi, f_\theta \leftarrow PPO(T)$
  **end while**

---

## 3 EXPERIMENTS

We ran over 300 experiments to thoroughly test our proposed contributions. Among our experiments, we varied network complexity, traffic flow dynamics, and state information availability to gauge the impact of each factor. The following sections outline some of our results.

### 3.1 SIMPLE NETWORK

Here, a simple environment, as shown in Fig. 4a, is used to train the *Transformer* model and the *Simple MLP* with all 3 different levels of available state information. The traffic flow demand within the network was kept static. Both models showed rapid convergence during training, and there was no significant difference

between model performances as seen in Fig. 4b (where *Simple MLP* is denoted as *MultiAgentPPO*). This result is hinting that – at least for simple road networks – limited observations are not a big issue.

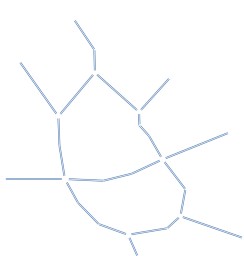

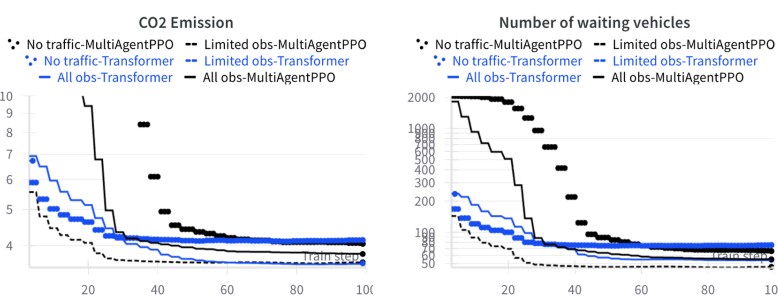

(a) A simple ring network with 7 agents.

(b) Training graphs for all models on the simple network. Note that multi-agent PPO corresponds to the baseline *Simple MLP*.

Figure 4: Training metrics for a simple network. All models converged with insignificant performance differences.

To evaluate our model, we compared it to the static traffic control baseline where the traffic signal phases are changed periodically irrespective of traffic flow information. The simulation ran for an hour, and as seen in Fig. 5, our model demonstrated improvements across all metrics: a 47% reduction in both fuel consumption and $CO_2$ emissions, and approximately a 90% decrease in the number of waiting vehicles. These results stem from the static traffic lights causing traffic jams in the simulation, whereas our model enhances traffic flow by reducing stop-and-go scenarios, thereby significantly improving these metrics.

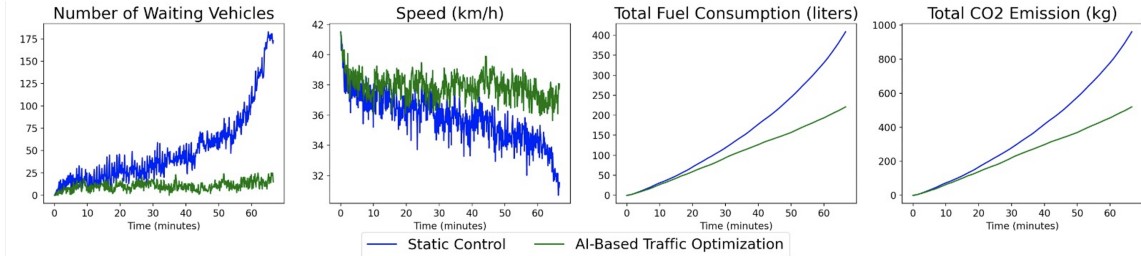

Figure 5: Evaluation of our transformer model against the static baseline.

## 3.2 COMPLEX NETWORK

In this experiment, a complex network is used to train our transformer model and additionally, the traffic flow demand within the network was made dynamic i.e. traffic flow varied with time. The model was trained on all levels of state information. Our model converged, regardless of the level of state information it was given, as seen in Fig. 6b.

## 3.3 MULTI-NETWORK

In subsequent experiments, we simultaneously trained our model on several road networks of differing complexities to develop a unified model capable of adapting and generalizing to diverse environments and

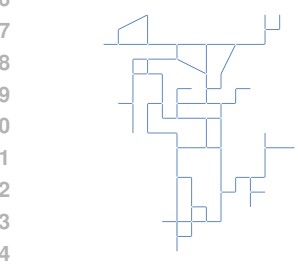

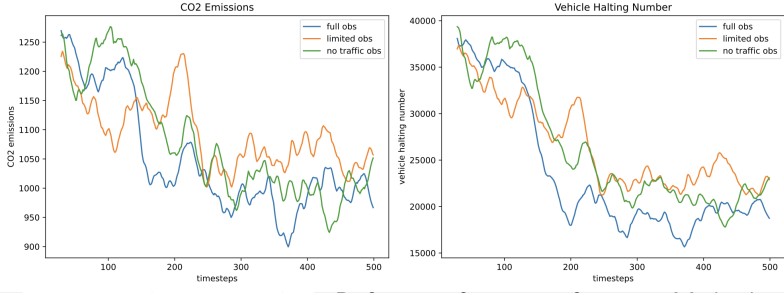

(a) A complex grid network with 73 agents.

(b) Training graphs show our model converges with all levels of state information.

Figure 6: Training metrics for complex network.

traffic demands. The traffic flow remained dynamic, and our transformer model was trained using all available levels of state information. Although our model has yet to show convergence with these advanced settings, the training pipeline is prepared for further experimentation and fine-tuning.

## 4  DISCUSSION & FUTURE WORK

For the simple network training experiments, as seen in section 3.1, the models showed rapid convergence and had similar performance during evaluation. Since this training was done with static traffic flows, it gave us our primary proof of concept for our novel approach to modeling the TSC problem: the transformer architecture was able to effectively model the communication using its attention mechanism without hindering training and convergence. Evaluation metrics in the simple network setup when compared to static baseline support this finding.

Further evaluation was done on more complex road networks. As outlined in section 3.2, our transformer model converged on all levels of state information, with similar performance on each level. It is an interesting finding that the performance does not seem to depend at all on the state knowledge, and all observation types reach the same results as seen in Fig. 6.

A possible avenue for future work, emerging from our multi-network environment experiments, is to develop a unified model capable of understanding diverse urban landscapes and traffic demands. This model aims to handle dynamic changes, such as construction sites, seamlessly across various city environments. Our contributed dataset, training pipeline and model architecture can serve as a strong foundation for this endeavor.

## 5  CONCLUSION

We were able to successfully engineer a training pipeline with randomized environment generation. Our novel architecture allows training on arbitrary environments without any modifications. Our approach to modeling this problem enables our model to handle variable sizes of input networks - essentially removing the need for training multiple networks based on network complexity.

Our results show the efficacy of using language modeling methods for the task of multi-agent RL for TSC. By modeling the agent communication as the primary sequence modeling problem, we showed that the agents were able to effectively communicate globally within the network.

Moreover, we were able to successfully show that limited state information to the agents can be sufficient to achieve competitive results. This is significant in reducing reliance on expensive sensor technologies to support convenient and cheap deployment to real-world road networks to reduce waiting times and $CO_2$ emissions in traffic globally.

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
