# OpenReview forum: "Large-Scale Multi-Agent Reinforcement Learning for Traffic Signal Optimization"
_ICLR.cc/2025/Conference — Submitted to ICLR 2025_

### Official Review · Reviewer_5gxv · 2024-10-31

**Soundness:** 2
**Presentation:** 1
**Contribution:** 2
**Rating:** 3
**Confidence:** 4

**Summary:**

This paper proposes a multi-agent RL method to enable communication between multiple intersections using sequence modeling. It also introduces a pipeline for dataset and road network generation. It further tries to handle issues with minimal state information.

**Strengths:**

The originality is good. It aims to solve some real-world problems in current traffic signal control research, such as varying network and intersection structures, multi-intersection coordination, and the lack of expensive sensor-captured data.

**Weaknesses:**

* The abstract doesn’t provide enough information about the problem, method, and contributions.
* Figures 5 and 6 both contain irrelevant parts in the screenshots. Moreover, the authors should use high-definition figures instead of screenshots.
* It’s not clear what the “difference in vehicle waiting time” means in the reward setting in Line 159.
* This paper states that one major contribution is the dataset, traffic flow and road network generation. However, how you generate them is not clearly explained.
* There are not any latest baseline methods for comparison.
* No uncertainty evaluation for any experimental results. Multiple runs are necessary for model evaluation.
* The experiments lack comprehensiveness, and the analysis does not provide sufficiently convincing insights.
* No code or implementation details are provided.

**Questions:**

In line 247, the Transformer model used in this paper assumes access to global information. As far as I know, information sharing is highly limited. Is global sharing reasonable in traffic signal control problems?

---

### Official Review · Reviewer_cjnA · 2024-11-01

**Soundness:** 1
**Presentation:** 1
**Contribution:** 2
**Rating:** 3
**Confidence:** 4

**Summary:**

This work proposes a novel RL framework for large scale traffic signal control, independent of the road network topology, based on the transformer architecture, together with a pipeline for generating diverse environments for this setting.

**Strengths:**

This work addresses an important and challenging setting, traffic signal control. It demonstrates the approach on large-scale settings, with up to 73 agents.

**Weaknesses:**

The work has merits and interesting contributions, but I argue it also has clarity issues and requires additional work on the empirical validation and presentation. I detail below some key issues and point to a few questions that can hopefully guide the future development of this contribution.

The first remark concerns the problem formulation. The multi-agent setting is defined using the MDP framework (i.e., single-agent setting), but using a joint action space. Should we understand that the MARL setting is approached using a centralised learning paradigm? Also the motivation for learning enriched states using information exchange signals a partially observable setting. I advise to reconsider the mathematical framework, given all these elements. Perhaps a Dec-POMDP [1] is more appropriate? See Q1.


Further clarity issues regards the two stage state encoding:
- The idea of using PointNet to generate an encoding independent of the road network size is interesting. But one can also argue that lane level information is detailed information, that is not always available. See Q3.
- It is not clear how exactly the communication is defined as a sequence modeling problem. As far as I understand the transformer further encodes into the state the topology information. See Q4.

Finally, the evaluation was only performed against MAPPO and it is unclear what the 300 mentioned experiments were, since the presented results do not seem to be averaged over multiple runs. There are numerous potential baselines that could strengthen the results: [2, 3, 4]. While the related work was great within the application domain, there is a lot left to explore on the algorithmic side.

[1] Oliehoek, F. A., & Amato, C. (2016). A concise introduction to decentralized POMDPs (Vol. 1). Cham, Switzerland: Springer International Publishing.

[2] Wen, M., Kuba, J., Lin, R., Zhang, W., Wen, Y., Wang, J., & Yang, Y. (2022). Multi-agent reinforcement learning is a sequence modeling problem. Advances in Neural Information Processing Systems, 35, 16509-16521.

[3] Jiang, J., Dun, C., Huang, T., & Lu, Z. Graph Convolutional Reinforcement Learning. In International Conference on Learning Representations 2020.

[4] Sheng, J., Wang, X., Jin, B., Yan, J., Li, W., Chang, T. H., ... & Zha, H. (2022). Learning structured communication for multi-agent reinforcement learning. Autonomous Agents and Multi-Agent Systems, 36(2), 50.

**Questions:**

Q1. Could you explain the MDP framework choice and why you did not opt for a multi-agent formalisation, such as a Dec-POMDP? What is the training and execution paradigm for your approach in this case? Is it a fully centralised approach?

Q2. Is the conditioning on the phase times and action space kept for the entire framework, I did not see any other reference to this beyond Eq. 3, or how this is incorporated in the policy network, after the augmented state is generated (Eq. 6, 7).

Q3. Is the lane level information a reasonable assumption in TSC settings?

Q4. Could you clarify how exactly the communication is formulated as a sequence modelling problem? Could you clarify what are the differences with [2] and would that approach be applicable in your setting? I would argue that might be an important related work to also discuss.

Q5. Can you motivate the baseline choice and why other GNN or communication/transformer-based MARL methods were not considered?

---

### Official Review · Reviewer_r7KQ · 2024-11-03

**Soundness:** 2
**Presentation:** 2
**Contribution:** 2
**Rating:** 3
**Confidence:** 4

**Summary:**

This paper proposes a Traffic Signal Control framework modeled as a multi-agent environment. The proposed framework consists of a Transformer architecture for feature extraction and an MLP for computing the action. The trained policy can be implemented for networks of different size.

**Strengths:**

The paper is well structured and was easy to follow. The problem considered in this manuscript is a very important problem and has been motivated very well in the introduction. The problem description and formulation is adequately. The use of transformer architecture is also well justified.

**Weaknesses:**

1) Clarity of writing and presentation: There are multiple instances where the paper lacks clarity in terms of writing and the meaning can only be understood with someone who are in this domain.

2) Novelty: The paper does not propose any innovative approach. The use of Transformer or Transformer type architectures with RL and MARL is a well-known approach. This paper showcases the implementation of the architecture for the TSC problem.

3) Baseline comparison: There should be a comparison to traditional TSC algorithms such as SCOOT. Without a comparison to traditional baseline methods, it is not possible to infer the effectiveness of the method.

4) The abstract can be more meaningful. The abstract should cover the outline of the paper. But that’s not the case here.

5) What is the 'static' baseline comparison? Please have a formal definition for it.

6) Figure 7b and 7c: it would be nice if a different color palette and markers were used to represent these plots. In its present for it is very difficult to interpret.

7)Font size for all the plots can be increased

**Questions:**

1) It is not clear how this problem can be formulated as a Markov Decision Process (MDP). In a real traffic flow situation, the state of the entire system is not affected by the action, but also by various other external factors.

2) How are collision and accidents handled? Are there negative rewards for this?

3)What is W_i in equation 4

---

### Official Review · Reviewer_yHgH · 2024-11-04

**Soundness:** 2
**Presentation:** 3
**Contribution:** 3
**Rating:** 5
**Confidence:** 3

**Summary:**

The paper presents a Novel Traffic Sign Control approach in a multi-agent setting by modelling communication as a sequence problem and allowing road networks to communicate. The model uses Graph Neural Networks to handle road network topologies and demonstrate that their approach can provide competitive performance despite minimal state information.

The paper presets:
- An automated pipeline for dataset generation

- Utilize a Transformer to model inter-agent dependencies by encoding state history.

- Capability to model non-uniform input size based on number of intersections and intersection sizes.

- A Model that can provide competitive performance to baselines without using expensive sensors.

**Strengths:**

- Experiments are done in both simple and complex network designs and show that ththe results are comparable to models that contain more information.

- Lane encoding is clearly presented in Figure 1.

- Utilizing existing Traffic Simulation models to verify the capabilities.

**Weaknesses:**

- It is unclear how the Graph Neural Network is used in this paper.

- Length of the Sequences for the provided results is not mentioned. It is unclear how the transformer based model would perform for longer sequences.

- Page limit is exceeded as per Lines 423 to 427.

**Questions:**

- How does the Communication model differ from Graph Based Multi-Agent Communication method presented in Seraj et al 2022 [1]?

- Can this model be used to inform the future design choices in urban transportation by iterative development and evaluation in order to increase the efficiency of the traffic flow?

[1] Seraj, Esmail, et al. _Learning Efficient Diverse Communication for Cooperative Heterogeneous Teaming_. No. SAND2022-4579C. Sandia National Lab.(SNL-NM), Albuquerque, NM (United States), 2022.

---

### Meta-Review · Area_Chair_acRP · 2024-12-29

**Metareview:**

The paper introduces Traffic Signal Control (TSC) in a multi-agent communication problem. They demonstrate the approach by training on real and randomly generated road networks with varying traffic demands, showing that competitive performance can be achieved even with minimal state information.

Reviewers acknowledge the importance and challenge of the traffic signal control setting, noting that the approach is tested on large-scale settings with many agents. However, several reviewers raise concerns about the clarity of presentation, and the novelty of the proposed method. Reviewers also question the formulation of the problem as a Markov Decision Process (MDP), suggesting that a Dec-POMDP might be more appropriate. Additionally, reviewers request a comparison to traditional TSC algorithms and other MARL methods, and request multiple runs. Overall, the reviewers generally agree that there are some interesting ideas, but that there are significant issues with clarity, novelty, and experimental validation that need to be addressed.

**Additional Comments On Reviewer Discussion:**

The authors did not provide a rebuttal.

---

### Decision · Program_Chairs · 2025-01-22

Reject